# Influence of pain on the quality of life in patients with venous ulcers: Cross-sectional association and correlation study in a brazilian primary health care lesions treatment center

Severino Azevedo de Oliveira Júnior[1], Adriana Catarina de Souza Oliveira[2]◯, Mayara Priscilla Dantas Araújo[1]◯, Bruno Araújo da Silva Dantas[3]◯*, Maria del Carmen García Sánchez[2]◯, Gilson de Vasconcelos Torres[4]◯

**1** Health Sciences Center, Federal University of Rio Grande do Norte, Natal, RN, Brazil, **2** Faculty of Nursing, Catholic University of Murcia, Murcia, Spain, **3** Faculty of Health Sciences of Trairi, Federal University of Rio Grande do Norte, Santa Cruz, RN, Brazil, **4** CNPQ (PQ1D), Health Sciences Center, Federal University of Rio Grande do Norte, Natal, RN, Brazil

◯ These authors contributed equally to this work.
* bruno.dantas@ufrn.br

**Data Availability Statement:** All data were deposited and published in the Mendeley Data

## Abstract

We aimed to verify the association and correlation between pain and QoL in people with VU treated in a Brazilian Primary Health Care (PHC) lesions treatment center. This is an observational, cross-sectional study with a quantitative approach, carried out in a service specialized in the treatment of chronic injuries, linked to 29 PHC units. Sociodemographic and health characterization instruments were used. The Short Form Health Survey-36 (SF-36) and Visual Analogue Pain Scale (VAPS) also were used. The Kruskal-Wallis test verified the association between the scalar variables of QoL and pain intensity. With Spearman's correlation test, we verified the level of correlation between the scales applied. A total of 103 patients participated in the study. Higher QoL scores associated with moderate pain were found, especially in the Physical role functioning, Physical functioning, and Vitality domains. Correlation analysis showed its greatest (moderate) strength in the interaction between the highest scores in the Physical role functioning and Emotional role functioning domains with the lowest pain levels.

## Introduction

Venous Ulcer (VU) is a lesion formed due to an underlying disease that affects the blood circulation of the veins and is one of the most frequent circulatory system diseases [1] and that occurs mainly in older people due to the evolution of Chronic Venous Insufficiency (CVI) [2]. This condition has a high prevalence, as observed in a study conducted in Australia, France, Germany, Italy, Spain, the United Kingdom and the United States, where approximately nine million cases were identified, being a condition with recurrence rates in 70% of patients [3]. In Brazil, a study found a prevalence of 2.9% of VU in elderly people treated in Primary Health

repository, available at https://data.mendeley.com/datasets/pnvsrf72kj.

**Funding:** This research was funded by the National Council for Scientific and Technological Development of Brazil awarded to the author and coordinator of the G.V.T. research through grant number 0257801662000850. The funders had no role in study design, data collection and analysis, decision to publish, or preparation of the manuscript.

**Competing interests:** The authors have declared that no competing interests exist.

Care (PHC). Since it has been shown to be associated with the clinical characteristics and life habits of the population [4].

The influence of socioeconomic profile on the incidence and severity of VU is also verified, especially in aspects such as female gender, advanced age and low education and income [5]. CVI-related diseases are more prevalent in Western countries, where more than 2% of the health budget is allocated for VU treatment [6]. However, a study conducted in Taiwan, located in the East, indicates that the weekly costs of treating a VU can reach almost US $300.00 (three hundred US dollars). In addition, it is known that treatment can last for several years until the lesion heals. Over time, patients experience several phases of disappointment due to the presence of the injury, experiencing depressive feelings, physical and social limitations, which directly impact on their Quality of Life (QoL) [6,7].

Especially when healthy habits are not present, VU is able to impact even more strongly on the QoL of the patients and family members [8], being a serious health problem with high demand for care [1]. In addition to these aspects, these injuries impact interpersonal coexistence, such as deprivation of freedom, alteration in self-image and self-esteem, lower productivity at work and presence of pain in its chronic form [5,8].

The pain, in turn, is often intense and can potentiate depressive symptoms, anxiety, sleep disorders and, consequently, impair QoL [9]. This implies higher health costs with drugs and materials for bandage [10]. In this sense, PHC is an essential tool for welcoming and adhering to VU treatment, requiring a holistic and humanized approach, with the inclusion of patients and family members in the treatment regimen with clarification of doubts and promotion of autonomy and their well-being [5].

It is understood that identifying the cause of pain, its mechanism of relief and its worsening may indicate strategies for its management more efficiently. Authors list pain as an important component of QoL, since the fact of feeling it represents for the individual a loss in their general well-being and their perception of good health [11]. Since it is known that pain is present in most people with VU, it is essential to care for wounds in a specialized and qualified way, by qualified and competent professionals. However, there are still few interventions that actually generate impactful results [12], such as the adoption of protocols with an expanded focus, aiming at the healing of lesions, such as the participation of patients and the context in which they are inserted. In the scenario of Brazil, some studies were published that point out the differences in QoL between people with VU, where it was observed that all its aspects were impaired [13,14]. However, little has been studied about the aspect of pain and its impact on the lives of these patients.

As a way of contributing to fill this gap, this study aimed to verify the association and correlation between pain and QoL in a Brazilian PHC lesions treatment center. Our primary hypothesis was that there was a negative correlation between aspects of QoL and pain in people with VU.

## Materials and methods

### Ethical aspects

The research was approved by the Research Ethics Committee of the Federal University of Rio Grande do Norte, Brazil, with opinion 156068, respecting the rights of human beings involved in research, according to the legislation in force in Brazil and according to what is provided for in the Declaration of Helsinki of 1975, revised in 2013, on good practices in clinical research. Before the application of any instrument, the objective of the study was explained and a signature was requested in the Informed Consent Form (ICF). The signature was made by the participant in writing.

## Study design and location

This is a descriptive, observational, cross-sectional study with a quantitative approach, which took place from August to October 2019, with patients treated in the PHC of the municipality of Parnamirim, State of Rio Grande do Norte, Brazil. The city has 29 PHC units distributed among the 23 neighborhoods of the city, which has a population of 272,490 inhabitants [15] and is located in the Metropolitan Region of Natal, capital of the state.

## Population and sample

The population was active patients with VU, that is, wounds open for at least three months without natural progression to the healing process, assisted in the research setting. A non-probabilistic sample was obtained for convenience.

Before data collection, each health unit was previously visited to obtain the total number of people with VU treated, which totaled 157 patients. Based on this quantity, the sample calculation was performed through an online formula of quantitative variables for finite populations (https://calculareconverter.com.br/calculo-amostral/). The population size, 95% confidence level and 5% margin of error were considered and 112 individuals were obtained. All units could be consulted and the final sample had representatives from all of them.

As inclusion criteria in the study, participants should be over 18 years of age; be registered in the local PHC; have at least one active VU below the knee and have an Ankle-Arm Index greater than 0.8 and less than 1.3. Exclusion criteria were having a fully healed VU of mixed or non-venous origin; having been discharged from treatment; and change of address to another city. To consider the VU active or cured, the presentation of the medical report that accompanied the patient during the treatment was considered. One hundred and three (103) people completed the study.

## Instruments and variables

The data collection script used in data collection were the sociodemographic and health characterization questionnaire, structured with closed and categorical questions, most of them divided into subcategories and formulated by the team of researchers themselves. It included six sociodemographic variables: gender, marital status, and profession/occupation (present or absent); level of education; age group and income. In addition to six health variables exploring the presence or absence of diseases: diabetes, hypertension, cardiovascular disease, CVI and smoking and daily sleep.

The *Short Form Health Survey* 36 (SF-36), an instrument used to measure QoL, with 36 questions on a Likert scale, directed to represent eight domains (Physical role functioning, Physical functioning, Pain, Social role functioning, Emotional role functioning, General health perceptions, Vitality, Mental health) and two dimensions: Physical Health and Mental Health [16]. The score ranges from 0 to 100, the higher the score the better the QoL [17]. This scale was translated and validated for Brazilian Portuguese [11]. The researchers preferred to choose the SF-36 over other instruments because they considered it more complete due to the number of aspects evaluated.

The Visual Analog Pain Scale (VAPS) was used to evaluate this aspect through a Likert scale for self-reported and subjective indication of pain, ranging from 0 to 10, with 0 being considered absence of pain, mild when from 1 to 3, moderate from 4 to 7 and severe from 8 to 10 (level 10 being the greatest pain that the individual has ever felt in life) [18]. However, for the treatment of the data of this study, the subcategories of pain intensity were created: Absent/Mild (0–3), Moderate (4–7) and Severe (8–10). Thus, the higher the score, the greater the intensity of pain.

## Data collection and availability

Data were collected in the form of face-to-face interviews during the consultations in the office of the service specialized in VU treatment and were guided by the questions of the selected instruments. Medical follow-up consultations and dressing changes took place weekly with each of the participants, according to the service's schedule. The researchers appeared on the days of the consultations without previously informing the participants and invited them to participate in the research. As it is a moment of exposure of the lesion, we seek to preserve the privacy of the participants, reducing the number of people on site and using physical barriers during collection. Before the research, participants were asked if they had used any type of pain reliever that day. If they answered in the affirmative, the collection would be scheduled for another time. However, none of the subjects reported that they used it. The researchers read aloud to the participants and presented the answer alternatives for their choice. The researchers were physicians, nurses, nutritionists, and physical therapists, all of whom were graduate students of master's and PhD degrees from the Federal University of Rio Grande do Norte, Brazil. Before carrying out the collections, all of them received training on the application of the instruments, as a qualification criterion to carry out the collection. Collections took place between August and October 2019, lasting approximately 60 minutes for each of them and under the supervision of the research coordinators. There was no blinding of any of those involved in the collections, and no remuneration or prizes were offered to participants or researchers. The authors did not have access to the personally identifiable information of the study participants during or after data collection.

## Data analysis and processing

The data were organized by the *Microsoft Excel software version 2021 (Microsoft Corporation, Washington, WA, USA)*, and exported to the *Statistical Package for the Social Sciences* (SPSS) statistical software version 22.0 (IBM, Armonk, NY, USA). For descriptive analyses, absolute and relative frequencies, percentiles, and median were used.

For nominal variables, we applied the normality test (Kolmogorov-Smirnov), identifying non-normal sample. For sample characterization, absolute ($n$) and relative (%) frequencies were used. Pearson's nonparametric chi-square test was used to evaluate the association between the characteristics of the sample and the presence of VU, and the percentiles with the domains and dimensions of the SF-36. The Kruskal-Walis test was used to evaluate the association between the domains and dimensions of the SF-36 instrument and pain scale, and the Spearman Rho test was used to evaluate the correlation ($r$) between these scalar pain variables (VAPS) and QoL (SF-36). In the correlation analysis, we adopted the following parameters to classify the correlation strength ($r$): weak <0.39, moderate between 0.40 and 0.50, and strong >0.50 [19]. To interpreting the results, we consider that a negative correlation (-) refers to the direct influence that the presence of pain exerts on the worst QoL. While the positive correlation (+) indicates that the presence of pain influences the best QoL. By Cronbach's α test, the SF-36 was evaluated with internal consistency of α = 0.78, in which values greater than 0.70 reveal high consistency [20]. For all tests, we considered the margin of error of 5%, Confidence Interval (CI) of 95% and the value of statistical significance when p<0.05 [19]. All data were deposited and published in the Mendeley Data repository, available at https://data.mendeley.com/datasets/pnvsrf72kj (accessed on 16 March 2023).

## Results

Although 157 individuals registered in the service were sought, 45 had moved to other locations. Thus, we obtained an initial sample of 112 participants. Of these, nine were excluded,

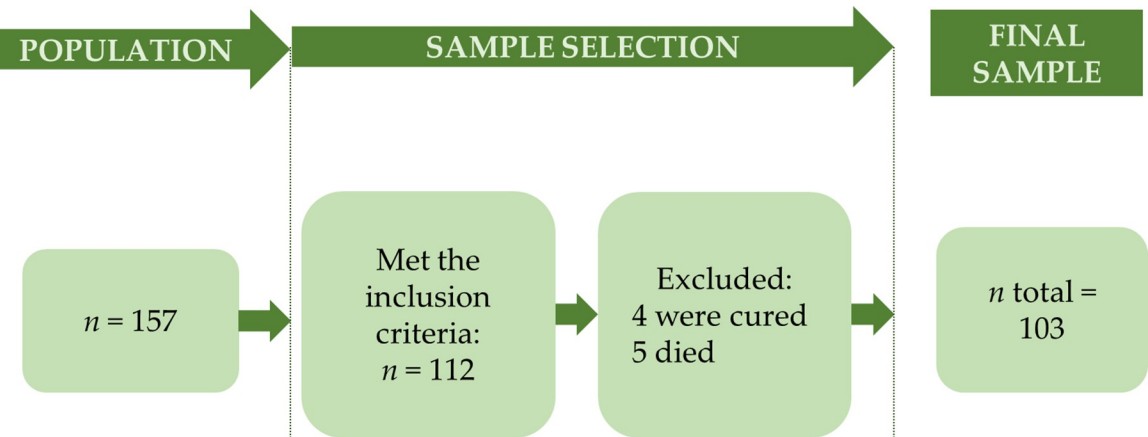

**Fig 1. Flowchart of the recruitment process and sample selection.**

four were discharged due to cure and five died in the time interval between the contact of the consultation and the interviews. Therefore, the final sample was n = 103. The sample selection process is shown in Fig 1.

Table 1 presents the data of the sociodemographic characteristics. There was a predominance of women, age ≥ 60 years, income of up to one minimum wage, not professionally active and education up to elementary school or less. Regarding health conditions, there was a predominance of people with arterial, with lesions in the malleolar region and foot, and with pain at the most severe level.

The behavior of the sample regarding the score obtained in the SF-36 is presented in Table 2 in a ranked manner according to its percentiles and mean. It is possible to observe that the domains General health perceptions, Mental health, Social role functioning and Vitality presented higher QoL scores significantly, while the Pain, Physical role functioning, Physical function, and Emotional role functioning domains exhibited lower QoL scores. Among the dimensions, there was no significance.

Table 3 presents the evaluation of QoL scores (SF-36) at each pain intensity level (VAPS). We found that in the General health perceptions, Vitality and Pain (p = 0.001) domains lower pain intensities were associated with higher QoL scores. The Physical role functioning, Physical functioning and Emotional role functioning domains exhibited an opposite behavior, so that lower pain levels were associated with lower QoL scores.

The findings of the correlation analysis between the aspects of QoL (SF-36) and pain intensity (VAPS) are presented in Table 4, in which they are ranked according to the correlation strength, from the negative to the positive direction. Among these data, the moderate and negative correlation found between the Physical role functioning and Emotional role functioning domains and pain intensity stood out. We also found the same level of correlation, but positive involving the pain and General health perceptions domains.

Additionally, we proceeded with the correlation analysis between the QoL variables within each of the groups of pain levels (Tables 5–7). The results were ranked according to the correlation strength (r) (from the strongest to the weakest between rows and columns) between the domains and dimensions, prioritizing those variables with statistical significance. In summary, we verified that the aspects that presented greater strength of correlation between them were the dimensions Mental Health and Physical Health, as well as the domains Physical role functioning, Physical functioning and Emotional role functioning.

**Table 1. Sociodemographic and health characterization of patients with venous ulcers.**

| Variables | Participants with VU (n = 103) | | p [a] |
|---|---|---|---|
| | n | % | |
| **Dados sociodemográficos** | | | |
| Gender | | | |
| Woman | 75 | 72.8 | <0.001 |
| Man | 28 | 27.2 | |
| Age group | | | |
| ≥60 years | 68 | 66 | 0.001 |
| <60 years | 35 | 34 | |
| Marital status | | | |
| Single/widowed/divorced | 53 | 51.5 | 0.768 |
| Married/stable union | 50 | 48.5 | |
| Income | | | |
| ≤1 minimum wage [b] | 85 | 82.5 | <0.001 |
| >1 and <3 minimum wage | 15 | 14.6 | |
| ≥3 minimum wage | 3 | 2.9 | |
| Ocupation | | | |
| Not active | 21 | 20.4 | <0.001 |
| Active | 82 | 79.6 | |
| Education | | | |
| Elementary school or less | 82 | 79.6 | <0.001 |
| High school or more | 21 | 20.4 | |
| **Living conditions and health** | | | |
| Diabetes | | | |
| Present | 33 | 32 | <0.001 |
| Absent | 70 | 68 | |
| Arterial hypertension | | | |
| Present | 62 | 60.2 | 0.039 |
| Absent | 41 | 39.8 | |
| Cardiovascular disease | | | |
| Present | 27 | 26.2 | <0.001 |
| Absent | 76 | 73.8 | |
| Alcoholism/ smoking | | | |
| Present | 9 | 8.7 | <0.001 |
| Absent | 94 | 91.3 | |
| Mental status | | | |
| Normal | 100 | 97.1 | <0.001 |
| Injured | 3 | 2.9 | |
| Lesion location | | | |
| Malleolar region/foot | 69 | 67 | 0.001 |
| Middle/upper third of the leg | 34 | 33 | |
| Medication [c] | | | |
| Use | 81 | 78.6 | <0.001 |
| Don't use | 22 | 21.4 | |
| Mobility | | | |
| Normal | 96 | 93.2 | <0.001 |
| Injured | 7 | 6.8 | |
| Pain | | | |

(*Continued*)

**Table 1.** (Continued)

| Variables | Participants with VU (n = 103) | | p [a] |
|---|---|---|---|
| | n | % | |
| Absent/Mild | 46 | 44.7 | <0.001 |
| Moderate | 9 | 8.7 | |
| Severe | 48 | 46.6 | |

VU: Venous Ulcer.

[a] Pearson's chi-square test.

[b] Minimum wage in Brazil in 2019: 1,100 BRL.

[c] Continuous use medication in general (e.g.: Antihypertensives, antidiabetics).

$\geq$: "greater than or equal to".

We suppressed the results of variables that did not exhibit moderate or strong correlation with statistical significance. But the complete table with all the data can be consulted in the S1 Table.

## Discussion

Our study brought as main evidence that better self-perceptions of QoL in relation to aspects of general health status and vitality were associated with lower pain intensities (absent or mild pain) in patients with VU. We also observed a greater negative correlation between aspects related to physical, functional and emotional state and pain intensity. That is, the participants showed better scores in these three aspects when lower levels of pain were observed, as well as their greater intensity was evidenced in lower scores of these aspects of QoL. This evidence fills an important scientific gap in the area of studies on patients with VU and allows professionals and managers to create health strategies and interventions in order to improve the QoL of these individuals, focusing on the weaknesses found, especially when the target population is inserted in the PHC and has similar sociodemographic characteristics.

**Table 2. Descriptive analysis of the scores of each aspect of QoL.**

| QoL (SF-36) (n = 103) | Percentiles | p [a] |
|---|---|---|
| | 25–50–75 | |
| Domains | | |
| General health perceptions | 50.0–70.0–75.0 | <0.001 |
| Mental health | 48.0–60.0–64.0 | <0.001 |
| Social role functioning | 50.0–50.0–62.5 | <0.001 |
| Vitality | 50.0–50.0–55.0 | <0.001 |
| Pain | 20.0–30.0–60.0 | 0.002 |
| Physical role functioning | 0.0–20.0–55.0 | <0.001 |
| Physical functioning | 0.0–0.0–75.0 | <0.001 |
| Emotional role functioning | 0.0–0.0–66.7 | <0.001 |
| Total score | 35.2–37.4–57.4 | 1.000 |
| Dimensions | | |
| Mental health | 43.0–47.9–60.0 | 1.000 |
| Physical health | 35.0–40.0–53.0 | 0.076 |

[a] Pearson's chi-square test.

**Table 3. Association between aspects of QoL (SF-36) and participants' pain levels (VAPS).**

| QoL (SF-36) | Pain levels (VAPS) (n = 103) | | | p [a] |
|---|---|---|---|---|
| | Absent/Mild (n = 46) | Moderate (n = 9) | Severe (n = 48) | |
| | Percentiles | | | |
| | 25–50[b] – 75 | 25–50[b] – 75 | 25–50[b] – 75 | |
| Domains | | | | |
| General health perceptions | 45.0–55.0–75.0 | 45.0–65.0–72.5 | 45.0–55.0–75.0 | <0.001 |
| Mental health | 48.0–56.0–64.0 | 44.0–64.0–68.0 | 48.0–60.0–64.0 | 0.556 |
| Social role functioning | 50.0–50.0–62.5 | 37.5–50.0–62.5 | 50.0–50.0–62.5 | 0.765 |
| Vitality | 50.0–50.0–56.2 | 55.0–55.0–65.0 | 45.0–50.0–55.0 | 0.028 |
| Pain | 37.5–50.0–60.0 | 15.0–20.0–40.0 | 10.0–20.0–55.0 | 0.001 |
| Physical role functioning | 0.0–5.0–36.2 | 2.5–40.0–62.5 | 6.2–45.0–65.0 | 0.002 |
| Physical functioning | 0.0–0.0–0.0 | 0.0–0.0–100.0 | 0.0–0.0–100.0 | 0.009 |
| Emotional role functioning | 0.0–0.0–0.0 | 0.0–0.0–100.0 | 0.0–0.0–100.0 | 0.005 |
| Total score | 35.1–37.3–42.0 | 32.5–41.3–61.8 | 35.7–40.1–59.9 | 0.276 |
| Dimensions | | | | |
| Mental health | 43.5–46.5–52.0 | 42.5–51.1–66.5 | 43.0–49.6–61.2 | 0.393 |
| Physical health | 35.0–39.0–45.0 | 34.5–40.0–57.0 | 34.3–41.0–54.0 | 0.729 |

[a] Kruskal-Wallis Test.

[b] Median.

Regarding the sociodemographic profile, the participants in our sample were mostly woman, aged 60 years or older (considered older people in Brazil), of low income and education, and did not have professional activity. These data are similar to other profiles found in the literature [5,21,22]. Authors explained that VU are more common in women, since in this public there is greater overload in the venous network, which presupposes fluid retention and

**Table 4. Correlation between the domains and dimensions of the QoL (SF-36) and pain (VAPS) in patients with venous ulcers.**

| Correlation between QoL (SF-36) and Pain (VAPS) (n = 103) | r [a] | p [b] |
|---|---|---|
| Domains | | |
| Physical role functioning | -0.34 | <0.001 |
| Emotional role functioning | -0.30 | 0.002 |
| Physical functioning | -0.28 | 0.004 |
| Social role functioning | -0.06 | 0.525 |
| Mental health | -0.05 | 0.643 |
| Vitality | 0.10 | 0.305 |
| Pain | 0.36 | <0.001 |
| General health perceptions | 0.40 | <0.001 |
| Total score | -0.16 | 0.115 |
| Dimensions | | |
| Physical health | -0.07 | 0.480 |
| Mental health | -0.10 | 0.324 |

[a] Spearman's Rho test.

[b] p-value for Spearman's coefficient.

Correlation levels: $r \leq 0.29$ (weak); $0.29 > r \leq 0.49$ (moderate); $r \geq 0.50$ (strong).

**Table 5. Correlation between QoL variables (SF-36) among participants with Absent/Mild pain (VAPS).**

| QoL (SF-36)—Pain Level Absent/Mild (n = 46) | QoL (n = 103) | | | | | | | | |
|---|---|---|---|---|---|---|---|---|---|
| | Mental Health Dimension | Physical health Dimension | Physical functioning | Emotional role functioning | Physical role functioning | Mental Health | General health perceptions | Vitality | Social role functioning |
| | $r^a$ ($p^b$) | r (p) | r (p) | r (p) | r (p) | r (p) | r (p) | r (p) | r (p) |
| Mental Health Dimension | - | 0.76 (<0.001) | 0.52 (<0.001) | 0.62 (<0.001) | 0.61 (<0.001) | 0.63 (<0.001) | 0.37 (0.011) | 0.35 (0.016) | 0.43 (0.003) |
| Physical functioning | 0.52 (<0.001) | 0.54 (<0.001) | - | 0.61 (<0.001) | 0.63 (<0.001) | 0.51 (<0.001) | -0.09 (0.955) | 0.09 (0.556) | 0.19 (0.192) |
| Physical health Dimension | 0.76 (<0.001) | - | 0.54 (<0.001) | 0.62 (<0.001) | 0.66 (<0.001) | 0.22 (0.130) | 0.63 (<0.001) | 0.17 (0.249) | 0.14 (0.332) |
| Emotional role functioning | 0.62 (<0.001) | 0.62 (<0.001) | 0.61 (<0.001) | - | 0.93 (<0.001) | 0.37 (0.010) | 0.19 (0.208) | 0.03 (0.808) | 0.12 (0.432) |
| Physical role functioning | 0.61 (<0.001) | 0.66 (<0.001) | 0.63 (<0.001) | 0.93 (<0.001) | - | 0.35 (0.016) | 0.22 (0.140) | 0.01 (0.942) | 0.11 (0.457) |
| Mental Health | 0.63 (<0.001) | 0.22 (0.130) | 0.51 (<0.001) | 0.37 (0.010) | 0.35 (0.016) | - | -0.08 (0.584) | 0.34 (0.019) | 0.19 (0.208) |
| General health perceptions | 0.37 (0.011) | 0.63 (<0.001) | -0.01 (0.955) | 0.19 (0.208) | 0.22 (0.140) | -0.08 (0.584) | - | -0.22 (0.130) | -0.23 (0.044) |
| Vitality | 0.35 (0.016) | 0.17 (0.249) | 0.09 (0.556) | 0.03 (0.808) | 0.01 (0.942) | 0.34 (0.019) | -0.22 (0.130) | - | 0.24 (0.110) |
| Social role functioning | 0.43 (0.003) | 0.14 (0.332) | 0.19 (0.192) | 0.12 (0.432) | 0.11 (0.457) | 0.19 (0.208) | -0.23 (0.044) | 0.24 (0.110) | - |

[a] Spearman's Rho test.

[b] p-value for Spearman's coefficient.

Correlation levels: $r \leq 0.29$ (weak); $0.29 > r \leq 0.49$ (moderate); $r \geq 0.50$ (strong).

greater difficulties in venous return when compared to men [1]. The advancement of age also applies in this context, but in addition to CVI with the effects of aging, there is a prevalence of chronic diseases in this group that increase the risk for the development of VU [23] and that were also present in our findings. The predominance of VU in older people may be due to the difficulty in pumping the venous valves in the calf region, resulting from the reduction of muscle strength and skin fragility, commonly observed in this population [23]. The incidence and prevalence of VU doubles after the age of 65, however, it is important to emphasize that they can also affect individuals in adulthood and that they require care at different stages of life [2].

**Table 6. Correlation between QoL variables (SF-36) among participants with Moderate pain (VAPS).**

| QoL (SF-36)—Pain Level Moderate (n = 9) | QoL (n = 103) | | | | |
|---|---|---|---|---|---|
| | Mental Health Dimension | Physical health Dimension | Physical functioning | Physical role functioning | Emotional role functioning |
| | r (p) | r (p) | r (p) | r (p) | r (p) |
| Physical functioning | 0.81 (0.009) | 0.78 (0.012) | - | 0.78 (0.012) | 0.78 (0.012) |
| Physical role functioning | 0.86 (0.003) | 0.87 (0.002) | 0.78 (0.012) | - | 1.000 (-)[c] |
| Physical health Dimension | 0.96 (<0.001) | - | 0.78 (0.012) | 0.87 (0.002) | 0.87 (0.002) |
| Mental Health Dimension | - | 0.96 (<0.001) | 0.81 (0.009) | 0.86 (0.003) | 0.86 (0.003) |
| Emotional role functioning | 0.86 (0.003) | 0.87 (0.002) | 0.78 (0.012) | 1.000 (-)[c] | - |

[a] Spearman's Rho test.

[b] p-value for Spearman's coefficient.

[c] p-value not shown in test due to maximum correlation strength.

Correlation levels: $r \leq 0.29$ (weak); $0.29 > r \leq 0.49$ (moderate); $r \geq 0.50$ (strong).

**Table 7. Correlation between QoL variables (SF-36) among participants with severe pain (VAPS).**

| QoL (SF-36)—Pain Level Severe (n = 48) | QoL (n = 103) | | | | | | | | |
|---|---|---|---|---|---|---|---|---|---|
| | Mental Health Dimension | Physical functioning | Physical role functioning | Emotional role functioning | Physical health Dimension | Mental Health | Vitality | General health perceptions | Pain |
| | r (p) | r (p) | r (p) | r (p) | r (p) | r (p) | r (p) | r (p) | r (p) |
| Mental Health Dimension | - | 0.51 (<0.001) | 0.76 (<0.001) | 0.83 (<0.001) | 0.69 (<0.001) | 0.64 (<0.001) | 0.33 (0.019) | 0.01 (0.931) | -0.63 (<0.001) |
| Physical role functioning | 0.76 (<0.001) | 0.70 (<0.001) | - | 0.91 (<0.001) | 0.82 (<0.001) | 0.40 (0.004) | -0.02 (0.859) | -0.36 (0.013) | -0.70 (<0.001) |
| Physical functioning | 0.51 (<0.001) | - | 0.70 (<0.001) | 0.61 (<0.001) | 0.72 (<0.001) | 0.43 (0.002) | 0.03 (0.812) | -0.39 (0.006) | -0.75 (<0.001) |
| Physical health Dimension | 0.69 (<0.001) | 0.72 (<0.001) | 0.82 (<0.001) | 0.73 (<0.001) | - | 0.26 (0.072) | -0.005 (0.970) | 0.03 (0.839) | -0.41 (0.003) |
| Emotional role functioning | 0.83 (<0.001) | 0.61 (<0.001) | 0.91 (<0.001) | - | 0.73 (<0.001) | 0.38 (0.008) | -0.01 (0.951) | 0.33 (0.022) | -0.68 (<0.001) |
| Mental Health | 0.64 (<0.001) | 0.43 (0.002) | 0.40 (0.004) | 0.38 (0.008) | 0.26 (0.072) | - | 0.57 (<0.001) | -0.13 (0.367) | -0.63 (<0.001) |
| Vitality | 0.33 (0.019) | 0.03 (0.812) | -0.02 (0.859) | -0.01 (0.951) | -0.005 (0.970) | 0.57 (<0.001) | - | 0.10 (0.507) | -0.19 (0.184) |
| General health perceptions | 0.013 (0.931) | -0.39 (0.006) | -0.36 (0.013) | -0.33 (0.022) | 0.03 (0.839) | -0.13 (0.367) | 0.10 (0.507) | - | 0.53 (<0.001) |
| Pain | -0.63 (<0.001) | -0.75 (<0.001) | -0.70 (<0.001) | -0.68 (<0.001) | -0.41 (0.003) | -0.63 (<0.001) | -0.19 (0.184) | 0.53 (<0.001) | - |

[a] Spearman's Rho test.

[b] p-value for Spearman's coefficient.

Correlation levels: $r \leq 0.29$ (weak); $0.29 > r \leq 0.49$ (moderate); $r \geq 0.50$ (strong).

Regarding education and income, authors point out that they can interfere with the development and progression of VU, since they determine the individual's access to medical and technological resources for effective treatment [24], as well as to better health education as a means of adopting healthy habits that prevent the worsening of the disease [25]. Another data that stood out was that most of the participants presented moderate intensity of pain. These findings are similar to a systematic review with meta-analysis, which indicated that 80% of people with VU live with pain of mild to moderate intensity [9]. In the case of these people, this may be related to local hypoxia triggered by poor circulation in the affected limb and which can cause physical mobility problems [26].

When performing the descriptive analysis of QoL variables (SF-36), we observed that the highest scores were in the domains General health perceptions, Mental health, Social role functioning, Vitality and pain. However, when we cross-referenced these data with the Pain variables (VAPS), we found that the highest QoL scores were significantly higher in the General health perceptions and Vitality domains. It should be noted that the mental and social aspects evaluated by the SF-36 relate to how the individuals feel in relation to their perception of mental health and their participation in social activities, such as visits to friends and family or meetings with other people that imply social interaction [11]. In patients with VU, these aspects gain prominence since the injury caused by the disease itself imposes on the individuals the coexistence with the aesthetic alteration of the affected limb and the fear of exposing themselves in environments with other people for fear of hurting them. Commonly, this context causes depressive thoughts in the person due to these limitations [13]. Pain, in turn, is a frequent and disabling symptom in these people that can also contribute to potentiate the problems of lack of socialization and reduction of mental health [27,28]. However, a comparative

study between elderly people from Brazil and Portugal, even without the presence of VU, observed that the participants showed an association between higher scores of the pain domain as the degree of depression increased [29]. Therefore, it is important to analyze the data considering that depressive symptoms may be related to other health or socioeconomic contexts.

The greater association between pain levels and the domains General health perceptions and Vitality was also a worrying finding. These domains are evaluated in the SF-36 by the individual's perception of how he sees himself in relation to his general health and his degree of energy and willingness to perform his daily activities [11]. It is known that pain is an important and representative symptom of how the individual feels about his general health, since it is a sensitive response to diseases and problems in the body, also interfering with his sleep quality and, consequently, his daily energy [5]. The pain caused by VU is considered chronic, which generates a continuous exposure to the various unpleasant sensations caused by it [28].

The Physical role functioning, Physical functioning, and Emotional role functioning domains, in turn, exhibited lower QoL scores when analyzed separately. Although they continue to show lower scores when crossed with pain scale, we observed that individuals with severe pain had higher QoL scores compared to those with absent or mild pain. Even so, correlation analysis demonstrated the highest coefficients for two of these three domains (Physical role functioning and Emotional role functioning), pointing to moderate and negative (inversely proportional) correlation with the pain scale (VAPS). In this sense, our findings also demonstrated that the Physical role functioning and Emotional role functioning domains were the only ones with a moderate and negative correlation level when crossed with the pain scale.

Regarding functionality and emotional health, the literature indicates a greater degree of impairment in these aspects in older people [29,30], age profile mostly found in our sample. However, it was not clear how much the presence of VU can interfere with QoL, as it is understood that there may be behavioral factors that can be altered, for example, by cultural issues, since this is already pointed out in other studies [13,31]. Another data that did not seem well understood was that, in the correlation analysis, the domains General health perceptions and pain showed their correlation as moderate, however in the positive sense between the scales. This finding seems to indicate that the better the QoL score in these aspects, there is a tendency for higher levels of pain.

Minimizing pain in VU treatment and improving QoL is essential [32]. Pain control actions can be medication or not. In addition to medication, other factors influence therapy, such as the correct orientation to the patient, professional experience, type of treatment for pain and the severity of VU. Non-pharmacological measures include lifestyle changes associated with elastic compressive therapy, with significant results in the healing of lower limb injuries and pain relief [33]. Effective treatment of VU with a focus on healing and improvement of QoL requires a long period of wound care. Some care actions are the responsibility of the PHC, such as the use of the appropriate dressing, the use of compressive therapy and pain control measures. In addition to the individual secondary care of each patient, such as adherence to the therapeutic regimen, weight control, practice of physical exercises, skin care and feeding, elevation of the lower limbs, among others. In addition, family support, elementary for recovery, support for carrying out daily activities and reintegration of the patient into the social environment [5]. It should be noted that although the participants in our sample are linked to PHC, all receive care in a specialized service for treatment and prevention of VU. Therefore, it is important to question whether our results would have any change if the participants did not receive this assistance.

In view of all that has been extracted from the literature, it is evident that VU affect the lives of the patients in the biopsychosocial, spiritual, economic aspects, and pain contributes to the reduction of QoL. It negatively impacts QoL regarding functional capacity, causing limitations

in physical aspects, pain, general health status, vitality, socioeconomic, emotional and mental health aspects [8]. Therefore, it is necessary to perceive the particularities about the presence of VU pain and its influence on QoL, enabling the construction of preventive and therapeutic measures that support the multiprofessional health team through the improvement of more effective public policies, aimed at this public, in the elaboration of therapeutic protocols pertinent to the reduction of pain and its side effects [34].

In addition, it is necessary to highlight the need for investments in PHC, as it is the main and initial sector of care for the prevention and recovery of various conditions. For this, more economic, physical and human resources are needed, especially professional training and health education of patients [35], in addition to providing continuous, humanized care based on sensitive listening, emotional and psychological support, integrating the family and establishing links between professionals and patients [8].

In general, our findings brought a diagnosis that pointed out the main aspects of QoL that are related to pain, such as the functional, physical and emotional aspects among the participants of our sample. This diagnosis may suggest the need for more specific interventions focused on these aspects, with a multiprofessional and continuous approach, which demonstrated success in other scenarios of the country that constitutes the present work (Brazil) [36].

## Limitation of study

As a limitation of our study, we did not reach the number of participants indicated by the sample test. We also asked about the use of daily medication by the participants, but we did not specifically investigate the use of psychoactive, which could compromise some interpretations of the study. Another important point is that the interviews were conducted during the consultations, which involved the VU exposure of the participants, which may have caused discomfort and triggered memories and feelings and, consequently, an overestimation in the answers provided. To reduce the possibilities of possible research biases, the data collection environments were reviewed and adapted in order to promote participants' privacy as well as a comfortable and welcoming environment.

## Conclusion

We concluded that there was a negative and moderate correlation between physical and emotional aspects and pain levels. We also found a moderate, however positive correlation between aspects of overall health and pain perception (SF-36) and pain levels. This seems to be a controversial fact. However, the association analysis showed that higher scores in these aspects were associated with lower pain intensity. These findings allowed us to confirm our study hypothesis.

Finally, we emphasize the need for a holistic, integrative and more efficient therapeutic approach in people with VU in PHC, resulting in greater autonomy and improved QoL. It is also suggested to improve the literature of randomized and multicenter studies focused on this area of knowledge, due to the difficulty in finding associations of QoL with the pain scale in people with VU.

## Supporting information

**S1 Table. Correlation between QoL variables (SF-36) among participants according to pain levels (VAPS).** [a] Spearman's Rho test; [b] p-value for Spearman's coefficient; [c] p-value not shown in test due to maximum correlation strength; Correlation levels: $r \leq 0.29$ (weak); $0.29 > r \leq 0.49$ (moderate); $r \geq 0.50$ (strong).
(DOCX)

## Acknowledgments

We especially thank all patients with venous ulcers who agreed to participate and contributed to the research.

## Author Contributions

**Conceptualization:** Severino Azevedo de Oliveira Júnior, Adriana Catarina de Souza Oliveira, Bruno Araújo da Silva Dantas.

**Data curation:** Gilson de Vasconcelos Torres.

**Formal analysis:** Gilson de Vasconcelos Torres.

**Funding acquisition:** Gilson de Vasconcelos Torres.

**Investigation:** Severino Azevedo de Oliveira Júnior, Adriana Catarina de Souza Oliveira, Mayara Priscilla Dantas Araújo, Bruno Araújo da Silva Dantas.

**Methodology:** Severino Azevedo de Oliveira Júnior, Adriana Catarina de Souza Oliveira, Mayara Priscilla Dantas Araújo, Maria del Carmen García Sánchez.

**Project administration:** Gilson de Vasconcelos Torres.

**Supervision:** Maria del Carmen García Sánchez, Gilson de Vasconcelos Torres.

**Validation:** Adriana Catarina de Souza Oliveira, Mayara Priscilla Dantas Araújo, Maria del Carmen García Sánchez.

**Visualization:** Maria del Carmen García Sánchez, Gilson de Vasconcelos Torres.

**Writing – original draft:** Severino Azevedo de Oliveira Júnior.

**Writing – review & editing:** Adriana Catarina de Souza Oliveira, Bruno Araújo da Silva Dantas, Maria del Carmen García Sánchez, Gilson de Vasconcelos Torres.

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
