## [Decision Letter · Decision Letter 0]

19 Jun 2023

PONE-D-23-10552Influence of pain on the quality of life in patients with venous ulcers: Cross-sectional association and correlation study in brazilian primary health carePLOS ONE

Dear Dr. Dantas,

Thank you for submitting your manuscript to PLOS ONE. After careful consideration, we feel that it has merit but does not fully meet PLOS ONE’s publication criteria as it currently stands. Therefore, we invite you to submit a revised version of the manuscript that addresses the points raised during the review process.

We look forward to receiving your revised manuscript.

Kind regards,

André Ricardo Ribas Freitas

Academic Editor

PLOS ONE

Journal Requirements:

https://www.mdpi.com/1660-4601/20/4/3583

In your revision ensure you cite all your sources (including your own works), and quote or rephrase any duplicated text outside the methods section. Further consideration is dependent on these concerns being addressed.

"This research was funded by the National Council for Scientific and Technological Development of Brazil awarded to the author and coordinator of the G.V.T. research through grant number 0257801662000850."

Additional Editor Comments:

Dear authors,

Thank you very much for your interest in publishing your study in our journal. This work contributes to knowledge about a very frequent health problem in primary care. The reviewers made some relevant suggestions that may contribute to the quality of this manuscript. I forward the contributions so that you can review your work if you agree.

Reviewers' comments:

Reviewer's Responses to Questions

**Comments to the Author**

1. Is the manuscript technically sound, and do the data support the conclusions?

Reviewer #1: Yes

Reviewer #2: Partly

2. Has the statistical analysis been performed appropriately and rigorously? 

Reviewer #1: Yes

Reviewer #2: No

3. Have the authors made all data underlying the findings in their manuscript fully available?

Reviewer #1: Yes

Reviewer #2: Yes

4. Is the manuscript presented in an intelligible fashion and written in standard English?

Reviewer #1: No

Reviewer #2: Yes

5. Review Comments to the Author

Reviewer #1: Comments to the Autor:

This is an interesting paper and will be of interest to those working in the área of chronic wounds and quality of life (QoL).

There a few points of clarification that would help the reader:

Materials and Methods

- Ethical aspects :

Please, explain in “Before the application of any instrument, the objective of the study was explained and a signature was requested in the Informed Consent Form” Did the patients signe the Informed Consent form?

- Instruments and variables:

Please, explain why did you use SF-36 and not a venous ulcer quality of life questionaire?

Did the patients take any analgesic?

- Data collection and availability

Please, explain how much time does it normally take from the contact with the patient to the interview?

Results

Please, In Tables 2 and 3, check the numbers are located in their rows.

Reviewer #2: A temática é atual, pertinente e interessante, pois, atualmente, observa-se um crescimento no diagnóstico das úlceras venosas (UV) e das condições que podem causá-las. O assunto já é bem explorado em estudos internacionais e já existem alguns estudos nacionais abordando a temática. Sugere-se que os autores destaquem os achados inéditos ou pouco explorados.

A leitura do manuscrito é interessante, entretanto, destaco os seguintes pontos:

Sugiro revisão no título do manuscrito, pois trata-se de um estudo pontual, realizado em um único centro, questiona-se se refletiria a realidade de toda a atenção primária brasileira, como consta no título.

O resumo apresenta os principais achados do manuscrito, entretanto, faço a mesma observação em relação ao título, questiona-se se um único estudo pontual poderia refletir a realidade brasileira?

A introdução é objetiva, interessante e chama a atenção para o problema, trazendo dados sobre a epidemiologia as UV. A justificativa para a realização do estudo está presente. O objetivo do estudo é descrito, porém, sugiro revisão em relação ao alcance do estudo (realidade brasileira?).

O método escolhido é adequado para se responder ao objetivo do estudo. A seção “Métodos” foi escrita de forma bem detalhada. Sugiro a utilização da instrução STROBE, como um guia para a elaboração do artigo.

Há informações repetidas sobre a população do estudo. Menciona-se a utilização de um “instrumento” para a coleta de dados sociodemográficos, porém, questiona-se se houve um processo formal de validação desse instrumento. Caso não seja validado, sugiro que seja utilizado o termo roteiro de coleta de dados.

Sugiro a inserção da referência adotada para a classificação das forças de correlação.

Questiona-se se o uso de medicamentos psicoativos foi investigado, pois seu uso poderia comprometer a investigação de constructos subjetivos, como a QV. A mesma questão pode estar relacionada ao diagnóstico de depressão.

Na seção “Resultados”, os autores repetem no corpo do texto, por várias vezes, os resultados apresentados nas tabelas, o que poderia tornar a leitura pouco fluida. Os resultados devem ser apresentados na tabela ou no texto, não em ambos.

Na tabela 1, os autores mencionam o status mental, bem como o uso de medicações, porém não fica claro seu significado ou quais medicações utilizadas foram investigadas. Sugiro que essas informações sejam detalhadas no texto subsequente à tabela.

A formatação das tabelas necessita de revisão.

Questiona-se se não seria mais adequada a apresentação das medianas nas tabelas ao invés das médias, pois foram utilizados testes não paramétricos. Além disso, os testes utilizados não permitem, por exemplo, verificar as diferenças inter/intragrupos entre os domínios do SF-36 e os participantes com dor ausente, moderada e forte. A utilização de uma análise estatística mais robusta poderia contribuir com o avanço do conhecimento e visibilidade do artigo.

Na seção “Discussão”, os autores trazem vários artigos, atuais e pertinentes, para fundamentar sua discussão, explorando as associações encontradas no estudo com a literatura disponível sobre o assunto, o que tornou a discussão ampla e interessante. Entretanto, questiona-se se realmente algumas das afirmações realizadas pelos autores são suportadas pelos resultados obtidos. Sugiro revisão.

As limitações são apresentadas ao final do estudo.

A conclusão responde ao objetivo do estudo.

Os autores apresentam 37 referências, sendo 25 dos últimos cinco anos.

6. PLOS authors have the option to publish the peer review history of their article (what does this mean?). If published, this will include your full peer review and any attached files.

Reviewer #1: No

Reviewer #2: No

---

## [Author Response · Author response to Decision Letter 0]

22 Jun 2023

Reviewer #1: 

Comments to the Autor:

This is an interesting paper and will be of interest to those working in the área of chronic wounds and quality of life (QoL). There a few points of clarification that would help the reader:

Materials and Methods

- Ethical aspects:

Please, explain in “Before the application of any instrument, the objective of the study was explained and a signature was requested in the Informed Consent Form” Did the patients signe the Informed Consent form?

Answer: The term was signed in writing by all participants. To make this clearer, we have added this information to the end of the description of the ethical aspects.

- Instruments and variables:

Please, explain why did you use SF-36 and not a venous ulcer quality of life questionaire?

Answer: We prefer to choose the SF-36 instead of other instruments because we consider it more complete due to the number of aspects evaluated. We add this information to the methodology in the body of the text.

Did the patients take any analgesic?

Answer: Before the survey, participants were asked if they had used any type of analgesic on the day of the interview. If they answered in the affirmative, the collection would be scheduled for another time. However, none of the individuals reported that they used it. We add this information in the body of the text in the topic "Data collection and availability". Thank you.

- Data collection and availability

Please, explain how much time does it normally take from the contact with the patient to the interview?

Answer: Medical follow-up appointments and dressing changes occurred weekly with each of the participants, according to the service's agenda. The researchers attended the days of the consultations without previously notifying the participants and invited them to participate in the research. This information was added to the text of the manuscript to make it clearer.

Results

Please, In Tables 2 and 3, check the numbers are located in their rows.

Answer: Some numbers were misaligned with the tables. We made the necessary adjustments. Thank you.

Reviewer #2:

A temática é atual, pertinente e interessante, pois, atualmente, observa-se um crescimento no diagnóstico das úlceras venosas (UV) e das condições que podem causá-las. O assunto já é bem explorado em estudos internacionais e já existem alguns estudos nacionais abordando a temática. Sugere-se que os autores destaquem os achados inéditos ou pouco explorados. 

A leitura do manuscrito é interessante, entretanto, destaco os seguintes pontos:

- Sugiro revisão no título do manuscrito, pois trata-se de um estudo pontual, realizado em um único centro, questiona-se se refletiria a realidade de toda a atenção primária brasileira, como consta no título. 

Answer: Realizamos a alteração sugerida delimitando o cenário para um centro de tratamento de lesões.

- O resumo apresenta os principais achados do manuscrito, entretanto, faço a mesma observação em relação ao título, questiona-se se um único estudo pontual poderia refletir a realidade brasileira? 

Answer: Fizemos uma alteração na redação direcionando o objetivo para a realização do estudo no centro de tratamento de lesões.

- A introdução é objetiva, interessante e chama a atenção para o problema, trazendo dados sobre a epidemiologia as UV. A justificativa para a realização do estudo está presente. O objetivo do estudo é descrito, porém, sugiro revisão em relação ao alcance do estudo (realidade brasileira?). 

Answer: Fizemos os ajustes a exemplo das observações anteriores.

-O método escolhido é adequado para se responder ao objetivo do estudo. A seção "Métodos" foi escrita de forma bem detalhada. Sugiro a utilização da instrução STROBE, como um guia para a elaboração do artigo. 

Answer: O recurso do checklist STROBE foi utilizado anteriormente e enviado preenchido junto à submissão inicial. No entanto, fizemos uma nova revisão diante de vosso comentário.

-Há informações repetidas sobre a população do estudo. 

Answer: Identificamos algumas informações que realmente estavam repetidas. Fizemos os ajustes necessários.

- Menciona-se a utilização de um "instrumento" para a coleta de dados sociodemográficos, porém, questiona-se se houve um processo formal de validação desse instrumento. Caso não seja validado, sugiro que seja utilizado o termo roteiro de coleta de dados. 

Answer: Como se trata de um texto escrito em inglês, substituímos pelo termo “data collection script”. 

- Sugiro a inserção da referência adotada para a classificação das forças de correlação.

Answer: A mesma referência que utilizamos para os demais parâmetros estatísticos foi a que escolhemos para a as forças de correlação. Para ficar mais claro, acrescentamos a mesma citação logo após a descrição dos parâmetros da força de correlação.

- Questiona-se se o uso de medicamentos psicoativos foi investigado, pois seu uso poderia comprometer a investigação de constructos subjetivos, como a QV. A mesma questão pode estar relacionada ao diagnóstico de depressão. 

Answer: De fato, o uso de psicoativos não foi investigado. Concordamos com a possibilidade de comprometer as interpretações. Fizemos um acréscimo sobre essa reflexão nas limitações do trabalho. No entanto, os participantes foram perguntados sobre o uso de medicamentos analgésicos, uma vez que o estudo se dedica a investigar os aspectos da dor. 

- Na seção "Resultados", os autores repetem no corpo do texto, por várias vezes, os resultados apresentados nas tabelas, o que poderia tornar a leitura pouco fluida. Os resultados devem ser apresentados na tabela ou no texto, não em ambos.

Answer: Retiramos todos os valores numéricos do texto que estão presentes na tabela.

- Na tabela 1, os autores mencionam o status mental, bem como o uso de medicações, porém não fica claro seu significado ou quais medicações utilizadas foram investigadas. Sugiro que essas informações sejam detalhadas no texto subsequente à tabela.

Answer: Essa variável se refere a medicações de uso diário em geral. O participante foi perguntado simplesmente se usa algum medicamento diariamente, para tratamento de alguma doença crônica, com hipertensão ou Diabetes. Acrescentamos uma legenda depois da tabela 1.

- A formatação das tabelas necessita de revisão. Questiona-se se não seria mais adequada a apresentação das medianas nas tabelas ao invés das médias, pois foram utilizados testes não paramétricos. Além disso, os testes utilizados não permitem, por exemplo, verificar as diferenças inter/intragrupos entre os domínios do SF-36 e os participantes com dor ausente, moderada e forte. A utilização de uma análise estatística mais robusta poderia contribuir com o avanço do conhecimento e visibilidade do artigo.

Answer: Fizemos os ajustes sugeridos para a formatação da Tabela 3. Nesse caso, não só mostramos a mediana representada pelo percentil 50, mas também a distribuição entre os três percentis, a fim de se padronizar o desenho com o das tabelas anteriores. Realizamos também uma análise adicional a fim de contemplar o apontamento da comparação inter/intragrupos. Foram acrescentadas as Tabelas 5, 6 e 7 que foram montadas individualmente (cada uma com um nível de intensidade da dor) e ranqueadas de acordo com a força de correlação e nível de significância entre as variáveis da qualidade de vida. A fim de melhorar a compreensão dos resultados, suprimimos algumas variáveis. No entanto, adicionamos como arquivo de suporte a Tabela S1 com todos os dados, sem supressão. Acreditamos que com essa análise, além de deixar nossos resultados mais robustos e esclarecedores, contemplamos integralmente vosso apontamento.

- Na seção "Discussão", os autores trazem vários artigos, atuais e pertinentes, para fundamentar sua discussão, explorando as associações encontradas no estudo com a literatura disponível sobre o assunto, o que tornou a discussão ampla e interessante. Entretanto, questiona-se se realmente algumas das afirmações realizadas pelos autores são suportadas pelos resultados obtidos. Sugiro revisão. 

Answer: Fizemos uma nova revisão e ajustamos a escrita de alguns fragmentos que poderiam ser interpretados equivocadamente como uma inferência de causa e efeito. Se ainda assim, perceber que o problema continua, peço gentilmente que aponte mais especificamente onde estão as afirmações que se encontram possivelmente sem a devida sustentação.

- As limitações são apresentadas ao final do estudo. A conclusão responde ao objetivo do estudo. Os autores apresentam 37 referências, sendo 25 dos últimos cinco anos.

Answer: Fizemos algumas alterações mediante solicitação dos revisores, assim como do editor.

---

## [Decision Letter · Decision Letter 1]

3 Aug 2023

Influence of pain on the quality of life in patients with venous ulcers: Cross-sectional association and correlation study in a brazilian primary health care lesions treatment center

PONE-D-23-10552R1

Dear Dr. Dantas,

We’re pleased to inform you that your manuscript has been judged scientifically suitable for publication and will be formally accepted for publication once it meets all outstanding technical requirements.

Kind regards,

André Ricardo Ribas Freitas

Academic Editor

PLOS ONE

Additional Editor Comments (optional):

Reviewers made minor suggestions that may be changed during the editing process.

Andre

Reviewers' comments:

Reviewer's Responses to Questions

**Comments to the Author**

1. If the authors have adequately addressed your comments raised in a previous round of review and you feel that this manuscript is now acceptable for publication, you may indicate that here to bypass the “Comments to the Author” section, enter your conflict of interest statement in the “Confidential to Editor” section, and submit your "Accept" recommendation.

Reviewer #1: All comments have been addressed

Reviewer #2: All comments have been addressed

2. Is the manuscript technically sound, and do the data support the conclusions?

Reviewer #1: Yes

Reviewer #2: Yes

3. Has the statistical analysis been performed appropriately and rigorously? 

Reviewer #1: Yes

Reviewer #2: Yes

4. Have the authors made all data underlying the findings in their manuscript fully available?

Reviewer #1: Yes

Reviewer #2: Yes

5. Is the manuscript presented in an intelligible fashion and written in standard English?

Reviewer #1: Yes

Reviewer #2: Yes

6. Review Comments to the Author

Reviewer #1: Tittle:

Please, check the number of words in the title.

I consider that the title is too long.

I think the number of words could be reduced.

Reviewer #2: Os autores abordaram corretamente os aspectos apresentados na revisão anterior. O artigo é atual, pertinente e interessante, contribuirá para o avanço do conhecimento.

Sugiro substituir no resumo o termo instrumento sociodemográfico e de saúde pelo mesmo termo mencionado na seção método.

As tabelas 5, 6 e 7 parecem necessitar de revisão quanto ao alinhamento. Sugiro que um padrão seja adotado, para facilitar a compreensão (os dados de uma célula em uma ou duas linhas em toda a tabela).

7. PLOS authors have the option to publish the peer review history of their article (what does this mean?). If published, this will include your full peer review and any attached files.

Reviewer #1: No

Reviewer #2: No

---

## [Editor Report · Acceptance letter]

7 Aug 2023

PONE-D-23-10552R1 

Influence of pain on the quality of life in patients with venous ulcers: Cross-sectional association and correlation study in a brazilian primary health care lesions treatment center 

Dear Dr. Dantas:

I'm pleased to inform you that your manuscript has been deemed suitable for publication in PLOS ONE. Congratulations! Your manuscript is now with our production department. 

Kind regards, 

on behalf of

Dr. André Ricardo Ribas Freitas 

Academic Editor

PLOS ONE